# Conducting the RBD of SARS-CoV-2 Omicron Variant with Phytoconstituents from *Euphorbia dendroides* to Repudiate the Binding of Spike Glycoprotein Using Computational Molecular Search and Simulation Approach

**DOI:** 10.3390/molecules27092929

**Published:** 2022-05-04

**Authors:** Heba Ali Hassan, Ahmed R. Hassan, Eslam A.R. Mohamed, Ahmad Al-Khdhairawi, Alaa Karkashan, Roba Attar, Khaled S. Allemailem, Waleed Al Abdulmonem, Kuniyoshi Shimizu, Iman A. M. Abdel-Rahman, Ahmed E. Allam

**Affiliations:** 1Department of Pharmacognosy, Faculty of Pharmacy, Sohag University, Sohag 82524, Egypt; heba.ali@pharm.sohag.edu.eg; 2Desert Research Center, Medicinal and Aromatic Plants Department, Cairo 11753, Egypt; drarahdrc@gmail.com; 3Department of Chemistry, Faculty of Science, Minia University, Minia 61511, Egypt; eslamahmedragabmohamed@gmail.com; 4Department of Biological Science and Biotechnology, Faculty of Science and Technology, Universiti Kebangsaan Malaysia, Bangi 43600, Selangor, Malaysia; ahmadayad@outlook.my; 5Department of Biology, College of Sciences, University of Jeddah, Jeddah 21959, Saudi Arabia; askarkashan@uj.edu.sa (A.K.); rmattar@uj.edu.sa (R.A.); 6Department of Medical Laboratories, College of Applied Medical Sciences, Qassim University, Buraydah 51452, Saudi Arabia; 7Department of Pathology, College of Medicine, Qassim University, Buraydah 51452, Saudi Arabia; dr.waleedmonem@qu.edu.sa; 8Department of Agro-Environmental Sciences, Graduate School of Bioresource and Bioenvironmental Sciences, Kyushu University, 744 Motooka, Nishi-ku, Fukuoka 819-0395, Japan; shimizu@agr.kyushu-u.ac.jp; 9Department of Pharmacognosy, Faculty of Pharmacy, South Valley University, Qena 83523, Egypt; emanabdelraheem@svu.edu.eg; 10Department of Pharmacognosy, Faculty of Pharmacy, Al-Azhar University, Assiut 71524, Egypt

**Keywords:** SARS-CoV-2 Omicron, *Euphorbia dendroides*, molecular docking, molecular dynamics

## Abstract

(1) Background: Natural constituents are still a preferred route for counteracting the outbreak of COVID-19. Essentially, flavonoids have been found to be among the most promising molecules identified as coronavirus inhibitors. Recently, a new SARS-CoV-2 B.1.1.529 variant has spread in many countries, which has raised awareness of the role of natural constituents in attempts to contribute to therapeutic protocols. (2) Methods: Using various chromatographic techniques, triterpenes (**1**–**7**), phenolics (**8**–**11**), and flavonoids (**12**–**17**) were isolated from *Euphorbia dendroides* and computationally screened against the receptor-binding domain (RBD) of the SARS-CoV-2 Omicron variant. As a first step, molecular docking calculations were performed for all investigated compounds. Promising compounds were subjected to molecular dynamics simulations (MD) for 200 ns, in addition to molecular mechanics Poisson–Boltzmann surface area calculations (MM/PBSA) to determine binding energy. (3) Results: MM/PBSA binding energy calculations showed that compound **14** (quercetin-3-*O*-β-D-glucuronopyranoside) and compound **15** (quercetin-3-*O*-glucuronide 6″-*O*-methyl ester) exhibited strong inhibition of Omicron, with Δ*G*_binding_ of −41.0 and −32.4 kcal/mol, respectively. Finally, drug likeness evaluations based on Lipinski’s rule of five also showed that the discovered compounds exhibited good oral bioavailability. (4) Conclusions: It is foreseeable that these results provide a novel intellectual contribution in light of the decreasing prevalence of SARS-CoV-2 B.1.1.529 and could be a good addition to the therapeutic protocol.

## 1. Introduction

In October 2021, a new SARS-CoV-2 B.1.1.529 variant emerged in South Africa, designated Omicron by the World Health Organization (WHO). By early 2022, over 100,000 Omicron genomes had evolved as Omicron begun to dominate SARS-CoV-2 infections around the world [1]. Because it contains more mutations than any other strain, it is more transmissible than previous strains. Many of the changes are found in the spike protein, which is involved in the transmission of the virus.

Natural ingredients offer a wide range of chemical properties, including antiviral activity, and thus could be used to treat coronavirus infections. Plants and their secondary metabolites that act against targets associated with SARS-CoV-2 infection could be useful leads for developing drugs against the newly emerged Omicron. The discovery of antiviral drugs and effective therapeutic techniques is lengthy and laborious. Therefore, natural chemicals are often considered as attractive alternative treatment solutions because they are also the main sources of antibacterial and antiviral drugs. Recently, it was suggested that dietary flavonoids could regulate the severity of SARS-CoV-2 disease by affecting ACE2 prevalence and function [2,3]. Several publications have also stated that polyphenols target the renin–angiotensin system by modulating angiotensin II levels in mice [4,5].

In silico studies recently reported that flavonoids such as quercetin and rutin and polyphenols such as epigallocatechin gallate, myricetin, and quercetagetin showed a high rate of inhibition against the RNA-dependent RNA polymerase (RdRp) of SARS-CoV-2 [6].

*Euphorbia dendroides* L. (family Euphorbiaceae), one of the species with high polyphenol content, is a perennial small tree growing in Sollum and Mersa Matruh in Egypt [7]. The genus Euphorbia is unique in that it includes highly reputed species used in traditional medicine against various human diseases, such as respiratory diseases, inflammation, skin diseases, diarrhea, migraine, gonorrhea, warts, and intestinal parasites and exerts a laxative effect [8,9,10]. In addition to *E. dendroides*, other plants in the genus Euphorbia and their constituents have also recently gained medicinal importance and are used for various diseases, including as anticancer [11,12,13], antioxidants [14], and antiviral agents, and to target multidrug resistance [15,16] and COVID-19 [17,18,19]. Our previous phytochemical studies on *E. dendroides’* aerial parts revealed the presence of ten phenolic compounds, including six flavonoids, one phenolaldehyde, and three phenolic acids [20], as well as six cycloartane triterpenes and lupeol triterpenes, in addition to β-sitosterol and three fatty acids [13]. The constituents of medicinal plants may serve as targets for the development of therapeutic candidates against some SARS-CoV-2 proteins [21,22]. Consequently, in this study, seventeen phytoconstituents from the categories of flavonoids, phenols, and triterpenes that we had previously obtained from this plant were evaluated by in silico studies as potential candidates against the Omicron variant of severe acute respiratory syndrome coronavirus-2 (SARS-CoV-2).

In this study, we performed computational molecular screening against the Omicron receptor-binding domain (O-RBD) to explore and develop drugs from natural plant constituents that are effective against the current pandemic virus.

## 2. Results and Discussion

### 2.1. Identification of Phytoconstituents from E. dendroides

The phytochemicals of *E. dendroides* (**1–17**) were elucidated using NMR (Appendix A) as well as LC-ESI-MS/MS spectra and by comparison with the literature data. These compounds were identified as 24-methylene cycloartan-3*β*-ol (**1**) [11], cycloart-23-ene-3*β*,25-diol (**2**) [23], cycloart-23-ene-3*β*,25-diol monoacetate (**3**) [23], 3*β*-hydroxy-cycloart-23-ene-25 methyl ether (**4**) [24], 24 *R*/*S*-3*β*-hydroxy-25-methylene cycloartan-24-ol (**5**) [24], 23 *R*/*S*-3*β*-hydroxycycloart-24-ene-23-methyl ether (**6**) [13], lupeol (**7**) [25], gallic acid (**8**) [26], vanillin (**9**) [27], protocatechuic acid (**10**) [28], *trans*-caffeic acid (**11**) [29], luteolin (**12**) [30], kaempferol-3-*O*-*β*-D-glucuronopyranoside (**13**), quercetin-3-*O*-*β*-D-glucuronopyranoside (**14**) [31], quercetin-3-*O*-glucuronide 6″-*O*-methyl ester (**15**) [31,32], kampferol-3-*O*-glucuronide 6″-*O*-methyl ester (**16**) [32], and quercetin-3-*O*-*β*-D-glucopyranoside (**17**) [33].

### 2.2. Molecular Docking

An important aspect of the drug discovery strategy is molecular docking analysis, which can be used to specify protein–ligand interactions in the active site of the target protein. In this context, all seventeen compounds were subjected to molecular docking calculations to investigate their potency as anti-Omicron drugs. For each compound, only the docking pose with the highest docking score was selected from the nine docking poses. The estimated docking scores for all seventeen compounds are shown in Appendix A. As shown in the data deposited in Appendix A, eight compounds had docking scores of less than −7.0 kcal/mol. For the remaining compounds, three compounds were in the range of −7.2 to −7.7 kcal/mol and six compounds were in the range of −7.9 to −8.8 kcal/mol. The average docking score for the seventeen compounds was calculated to be −7.04 kcal/mol.

Remdesivir is a nucleotide analog prodrug and has been recently subjected to in vitro experiments as an anti-Omicron drug [34]. In order to assess the potentiality of the discovered compounds, the binding features and affinities of the top six compounds were compared to those of remdesivir against Omicron. Table 1 provides a deeper understanding of the binding characteristics for the top six compounds, as well as remdesivir, with Omicron.

The docking features in Table 1 reveal that remdesivir demonstrated four hydrogen bonds with TYR453, SER494, and TYR501 with bond lengths ranging from 2.27 to 3.01 Å. The six top-ranked compounds exhibited similar binding modes and abundant hydrogen bonds with three main residues: SER496, TYR501, and HIS505. For instance, compound **15** achieved the highest value of docking score towards Omicron equal to −8.8 kcal/mol, forming multiple hydrogen bonds with SER496 (2.99, 3.01 Å), in addition to TYR501 (2.92 Å) and HIS505 (3.17 Å). Compound **14**, the second-highest-ranked compound, had a docking score equal to −8.7 kcal/mol, forming triple hydrogen bonds with SER496 (2.13, 2.93, 3.01 Å), as well a single bond with TYR501 (2.89 Å) and HIS505 (3.18 Å).

To recognize the other types of interactions, two-dimensional representations of the interactions of these six potent compounds with the main active site residues are displayed in Figure 1. It is worth mentioning that compound **15** exhibited a pi–alkyl interaction with LEU455 (4.96 Å) and ARG493 (4.17 Å). Moreover, it also formed a pi–cation interaction with ARG403 (4.92 Å), as well pi–pi stacked and pi–pi T-shaped with TYR501 (4.24 Å) and HIS505 (4.92, 4.99 Å), respectively. In the case of compound **14**, it formed two extra types of interactions, pi–pi T-shaped with HIS505 (4.92, 5.05 Å) and a pi–cation interaction with ARG403 (4.90 Å). Similar to compound **15**, compound **16** formed pi–alkyl, pi–pi stacked, and pi–cation interactions with the same residues but different bond lengths. Excluding pi–alkyl interaction, compound **17** showed the same types of interactions that existed in compound **16**. Notably, compound **13** was the only one that exhibited no pi–cation interaction. Compound **12**, the lowest-ranking of the six compounds in terms of docking score, showed only two additional interactions other than hydrogen bonding. It is worth highlighting that residues ARG403, TYR501, and HIS505 favored the formation of pi–cation, pi–pi stacked, and pi–pi T-shaped interactions, respectively, with RBD residues.

### 2.3. Molecular Dynamics (MD) Simulations

The main purpose of applying molecular dynamics (MD) simulations is to investigate the conformational flexibilities and stabilities of studied protein–ligand complexes. Accordingly, MD simulations up to 200 ns were performed for the best-ranking six complexes. Binding energy calculations were also run using the MM/PBSA approach. Data of estimated binding energies are represented in Figure 2.

Notably, compounds **14** and **13** showed similar binding energies over 50 ns with values equal to −44.8 and −46.4 kcal/mol, respectively. Furthermore, compound **15** and compound **16** showed identical values of binding energy. Interestingly, compound **12**, the lowest of the six compounds in terms of docking score, also exhibited the lowest binding energy, with a value equal to −18.6 kcal/mol.

To further check the stability of the six compounds inside the Omicron active site, MD simulation was performed up to 100 ns. Remarkably, compounds **15** and **12** exhibited higher binding energies than in the 50 ns MD simulations, with Δ*G_binding_* of −22.9 and −21.8 kcal/mol, respectively. Compounds **14** and **13**, those that were the highest-ranked compounds in the 50 ns MD simulations, were also the highest-ranked in the 100 ns MD simulations, with a slight decrease in binding energies, with values equal to −42.4 and −38.4 kcal/mol.

Through analyzing the binding energies for the six compounds over the whole 200 ns MD simulations, compound **13** showed a continuous decrease in binding energy with increasing simulation time. Conversely, compound **15** exhibited increasing binding energy over the 200 ns MD simulations. Interestingly, compound **17** showed very similar results of binding energy over the 50 ns, 100 ns, 150 ns, and 200 ns MD simulations.

### 2.4. Post-MD Analyses

The structural stability and conformational variations of protein–ligand docked complexes over the MD simulation process can be evaluated by root-mean-square deviation (RMSD) measurement. Lower values of RMSD give an indication of tight binding. The following equation describes how RMSD was estimated.
$$\mathrm{RMSD}=\sqrt{\frac{\sum_{i=1}^{n}R_{i}*R_{i}}{n}}$$
where *R_i_* is the vector connecting the positions of atom *i* [of N atoms] in the reference snapshot and the current snapshot after optimal superposition.

Obtained results of RMSD analysis over 200 ns MD simulation are plotted in Figure 3. It is worth mentioning that compound **14** exhibited overall stability compared to the other five compounds, with an average RMSD value of 2.84 Å. Comparatively, the RMSD values of the other compounds were relatively high. These results showed that compound **14** was tightly bound and had no effect on the overall topology of Omicron.

Root-mean-square fluctuation (RMSF) analysis was conducted to investigate the flexibility of the Omicron residues over the 200 ns MD simulations. Briefly, higher RMSF values indicate greater flexibility of protein residues, whereas low RMSF values imply limitations of residues’ movement, and accordingly less flexibility. RMSF can be estimated using the following equation:$$\mathrm{RMSF}=\sqrt{\overset{3}{\underset{j=1}{\sum }}(\frac{1}{N}\overset{N}{\underset{k=1}{\sum }}P_{ijk}^{2}-{\bar{P}}_{ij}^{2})}$$

The RMSF of the atom *i* with *j* from 1 to 3 for the x, y, and z coordinate of the position vector *P* of the atom and *k* over the set of *N* evaluated snapshots was calculated.

The RMSF data are shown in Figure 4. Compound **14** complexed with Omicron had fewer fluctuations over 200 ns, with RMSF of 1.47 Å, which is consistent with the RMSD findings. Other compounds had values of RMSF ranging between 1.48 and 1.68 Å.

In order to assess whether the protein–ligand complexes were stably folded or not, radius of gyration (Rg) analysis was performed over the 200 ns MD simulations. Values of Rg give an indication of the compactness of the protein structure within the system [35] A more compact protein structure can be observed in the case of lower values of Rg. During the 200 ns simulation course, all compounds exhibited acceptable behavior of Rg, which can be observed in Figure 5. The range of Rg values for the six compounds was from 18.68 to 19.14 Å. What also can be noticed from Figure 5 is that compounds **14** and **15** showed constant Rg behavior during the simulation, which indicates high compactness of the protein structure.

Solvent-accessible surface area analysis (SASA) was performed to represent the area of protein exposed to solvent. The averaged SASA values of the six structures were 10,501, 10,428, 106,30, 11,007, 10,553, and 10,886 for compounds **12** to **17**, respectively. Despite the compact folding of 7QNW, the increase in SASA in compound **15** compared to the other compounds indicated obvious conformational changes due to ligand binding. Consequently, SASA may provide details about the protein’s ability to interact. As shown in the data in Figure 6, all six compounds exhibited relative stability and compactness over the course of the 200 ns MD simulation.

### 2.5. In Silico Drug Likeness

Based on Lipinski’s rules, physicochemical properties were investigated to understand the studied compounds’ molecular features better. The Molinspiration tool https://www.molinspiration.com (30 April 2022) was utilized to compute the in silico molecular features of compounds. The investigated Lipinski’s parameters, topological polar surface area (TPSA), as well percentage of absorption (% ABS) of Lipinski’s parameters were anticipated and are presented in Table 2.

As revealed from the data in Table 2, the investigated two compounds showed lower values of miLogP, indicating that these compounds possess adequate permeability via the cell membrane. The molecular weights of the inspected compounds did not exceed 500 (calc. 478.4 and 492.4). Hydrogen bond donors (nON) were found to total 13. Furthermore, the number of hydrogen bond acceptors (nOHNH) was found to range between 7 and 8. Although the values of nON and nOHNH were higher than the ideal values, it was reported that this defect did not exert a remarkable effect on the compound’s diffusion and transportation, as many FDA-approved drugs transcend the optimum Lipinski values of nON and nOHNH [36]. The estimated %ABS values were in the range of 30.0–35.0%. The TPSA values were also detected in the range of 215.0–230.0, which indicates the high bioavailability of these discovered compounds.

## 3. Materials and Methods

### 3.1. Plant Material

The aerial parts of *E. dendroides* were collected in April 2017 in Mersa Matruh on the northwestern coast of Egypt. The plant material was authenticated by Dr. Omran Ghaly, with a PhD in Plant Taxonomy, at the Desert Research Center. A voucher sample (CAIH-30-12-2017-R) was deposited in the herbarium of the Desert Research Center, Cairo, Egypt.

### 3.2. Phytochemical Constituents of E. dendroides

The phytochemical study of the aerial plant parts of *E. dendroides* revealed approximately seventeen secondary metabolites, as indicated by previous studies [13,20]. These natural components have been divided into six flavonoids (**12**–**17**), four phenolic compounds (**8**–**11**), and seven triterpenes (**1**–**7**). In particular, the triterpenes were extracted from the methanol plant extract by extensive chromatographic methods, and their structure was elucidated by 1D and 2D NMR spectroscopic methods. Four phenolics and three flavonoids were also isolated and identified from the poly-phenolic-rich fraction of the plant using chromatographic and spectroscopic tools, while the remaining three flavonoids were determined from the polyphenol-rich fraction of the plant using LC-ESI-MS/MS. All the phytochemical components of *E. dendroides* (**1**–**17**) were tested in silico for their ability to inhibit Omicron.

### 3.3. Protein Preparation

The three-dimensional crystal structure of the receptor-binding domain (RBD) of the SARS-CoV-2 Omicron variant (PDB ID: 7QNW, resolution: 2.40 Å) was retrieved and used for all in silico analyses. The downloaded viral target was prepared by removing ions, water molecules, and hetero-atoms. To identify the protonation states of the protein residues, the H++ web server [37,38] was used. Accordingly, all missing hydrogen atoms were successfully added. For the H++ calculations, physiological parameters such as pH, salinity, internal dielectricity, and external dielectricity were set to 6.5, 0.15, 10, and 80, respectively.

### 3.4. Inhibitor Preparation

Chem3D Pro 12.0 software (version 12.0.2) was utilized to sketch and analyze the seventeen extracted compounds’ chemical structures. All studied compounds were subjected to energy minimization using the MM2 force field. Before a molecular docking study, such an energy minimization step is required to reduce the influence of any potential unfavorable torsion angles, bond angles, bond lengths, or undesirable non-bonded interactions [39]. The names of the investigated compounds and their 2D chemical structures are illustrated in Appendix A.

### 3.5. Molecular Docking

Molecular docking is considered the best tool in computational drug discovery to determine the efficacy of the compounds under study [40]. AutoDock Vina was used to study the binding affinities for these compounds [41]. All parameters were left in their default modes in this study, except for the exhaustiveness parameter, which was set to 200. Residues of the O-RBD were enclosed by a docking grid box with XYZ dimensions of 25 × 25 × 25 (Å). In addition, the grid spacing was set to 1.0 Å. The generated nine poses of the docked inhibitors were evaluated and the best one was selected. BIOVIA Discovery Studio was used to visualize the protein–ligand interactions [42].

### 3.6. Molecular Dynamics Simulations

Molecular dynamics (MD) simulations were conducted using the YASARA Structure (version 21.12.19) protocol [43] to obtain a better understanding of the stability of the protein–ligand complexes. Within the MD simulations, the AMBER14 force field was the utilized force field. Execution of the initial energy minimization was performed using the steepest descent algorithm. The MD simulations were conducted for amino acid residues at the default physiological value of pH (7.4). Water molecules were successfully introduced into the system at constant temperature and pressure conditions. Counter ions (Na^+^ or Cl^−^) with a concentration equal to 0.9% were included to maintain the neutral state of the systems. To maintain the pressure value at 1 atm, the Berendsen barostat technique [44] was used. The long-range coulomb forces were computed employing the particle-mesh Ewald (PME) method [45,46]. The cut-off radius was set to 8 Å for the non-bonded interactions. The Langevin thermostat method was employed to hold the value of temperature at 300 K [47]. The periodic boundary conditions were also taken into account. The cubic simulation cell was chosen to be larger than the studied protein–ligand complexes by 20 Å in every instance. With a multiple time step of 1.25 fs, a regular simulation speed was preserved for intramolecular processes. At an integration step of 2 fs, all intermolecular bonds, including hydrogen bonds, were constrained using the SHAKE algorithm [48]. As a final step, production stages were accomplished over simulation times of 50 ns, 100 ns, 150 ns, and 200 ns. Snapshots of the simulation trajectory were held every 100 ps after an equilibration time of 1–2 ns, determined by the root-mean-square deviations (RMSDs) of the solutes from the initial structure. Simulation steps were executed using a pre-installed macro (md_run.mcr) within the YASARA package. The YASARA program (version 21.11.16) uses the Poisson–Boltzmann approach [49] (called “PBS”). The surfcost parameter, used to calculate the entropic cost of exposing an Å^2^ to the solvent, was set to 0.35. The Amber14 force field was used to calculate the binding energy of the inhibitor. To ensure consistency with the empirically determined values, the binding energies derived from PBS were divided by a factor of 20 [50]. The root-mean-square deviation (RMSD), radius of gyration (Rg), root-mean-square fluctuation (RMSF), and solvent-accessible-surface-area (SASA) were all used to analyze the best compounds at the end of the 200 ns MD simulations.

### 3.7. Binding Energy Calculations

Binding free energies of the investigated drugs against Omicron were computed using the molecular mechanics Poisson–Boltzmann surface area (MM/PBSA) approach [51]. The below-illustrated equations were used in the process of MM/PBSA binding free energy calculations.
$$\Delta G_{binding}=\;\Delta G_{C}-\Delta G_{P}-\;\Delta G_{L}$$
$$\Delta G_{binding}=\Delta H-T\Delta S=\Delta E_{MM}+\Delta G_{Sol}-T\Delta S$$

Values of the binding energy of the complex, protein, and ligand are described by Δ*G_C_*, Δ*G_P_*, and Δ*G_L_*, respectively. In addition, Δ*G_Sol_*, Δ*E_MM_*, and −*T*Δ*S* stand for the solvation Gibbs energy, gas-phase molecular mechanics change, and conformational entropy, respectively. The term Δ*E_MM_* can be determined through summation of the van der Waals and electrostatic interactions. The term Δ*G_Sol_* can readily be defined as adding the polar and non-polar solvation values. The entropic contribution is denoted by the term −*T*∆*S*.

### 3.8. Drug Likeness Properties

In order to assess the physicochemical parameters of the specified compounds, the online Molinspiration cheminformatics software (http://www.molinspiration.com 30 April 2022) was employed. For each discovered compound, various descriptors were checked, including the octanol/water partition coefficient (milogP), topological polar surface area (TPSA), molecular weight (MWt), number of hydrogen bond donors (nOHNH), number of hydrogen bond acceptors (nON), number of rotatable bonds (Nrotb), and percentage of absorption (%ABS). The equation that was used in computing %ABS is displayed below [52]:%ABS=109−[0.345×TPSA]

## 4. Conclusions

In the current pandemic, with the emergence of Omicron, in silico strategies may be useful in discovering potent inhibitors for this disease. In the present study, molecular docking calculations were performed for seventeen isolated compounds from *Euphorbia dendroides*. The results showed that six flavonoid compounds are the best anti-Omicron drug candidates, further supporting their efficacy. Combined molecular dynamics simulations (MD) and molecular mechanics Poisson–Boltzmann surface area (MM/PBSA) binding energy evaluations over 200 ns were performed for these six compounds. Two compounds, namely quercetin-3-O-β-D-glucuronopyranoside and quercetin-3-O-glucuronide 6″-O-methyl ester, showed promising binding affinities, with Δ*G_binding_* of −41.0 and −32.4 kcal/mol, respectively. As for drug-like properties, both compounds also proved their potential, with a good percentage of absorption (%ABS).

## Figures and Tables

**Figure 1 molecules-27-02929-f001:**
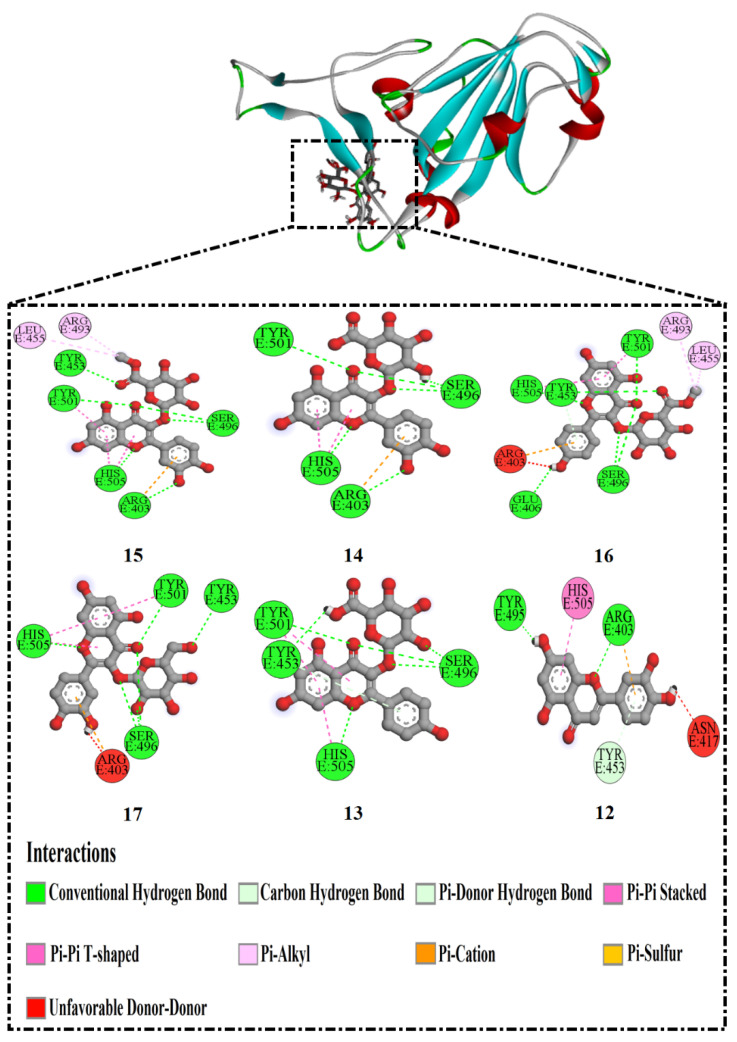
Two-dimensional representations of the anticipated binding poses of the best-investigated drugs inside the active site of Omicron.

**Figure 2 molecules-27-02929-f002:**
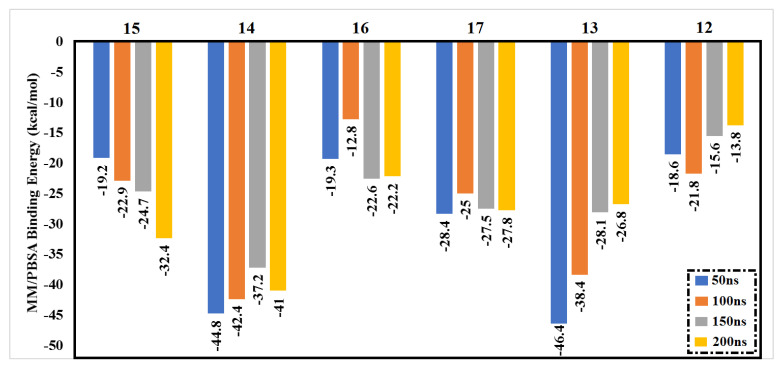
Estimated MM/PBSA binding energies for the best six drug candidates as Omicron inhibitors.

**Figure 3 molecules-27-02929-f003:**
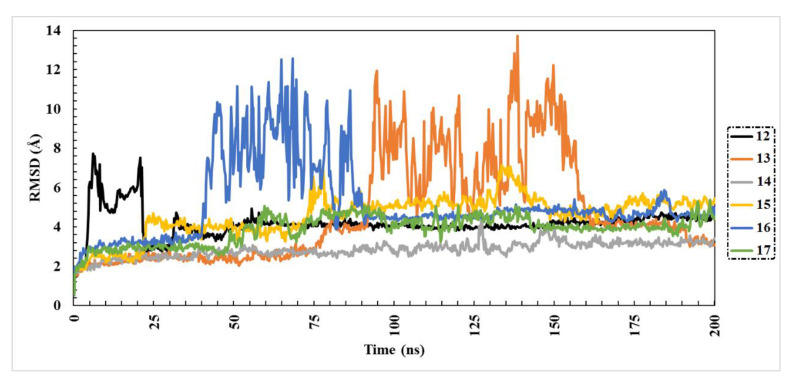
Root-mean-square deviation (RMSD) of Omicron backbone atoms from the initial structure complexed with the highest-ranked drugs over 200 ns MD simulations.

**Figure 4 molecules-27-02929-f004:**
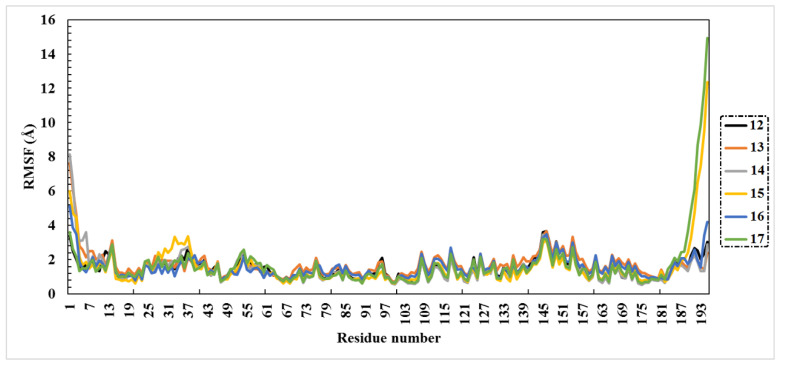
Root-mean-square fluctuation (RMSF) of the six selected complexes (Omicron-**12**, Omicron-**13**, Omicron-**14**, Omicron-**15**, Omicron-**16**, Omicron-**17**).

**Figure 5 molecules-27-02929-f005:**
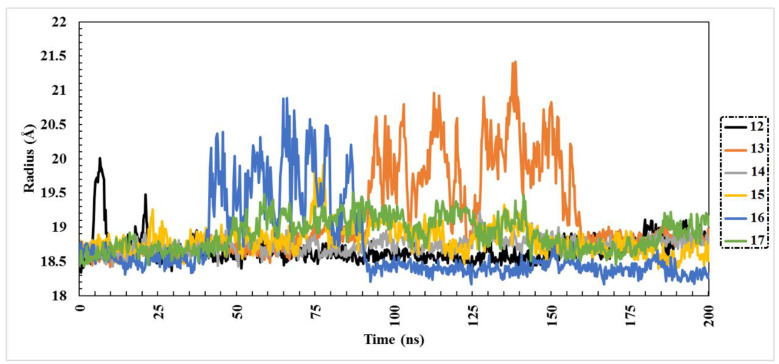
Radius of gyration (Rg) plot of six identified complexes through 200 ns MD simulations.

**Figure 6 molecules-27-02929-f006:**
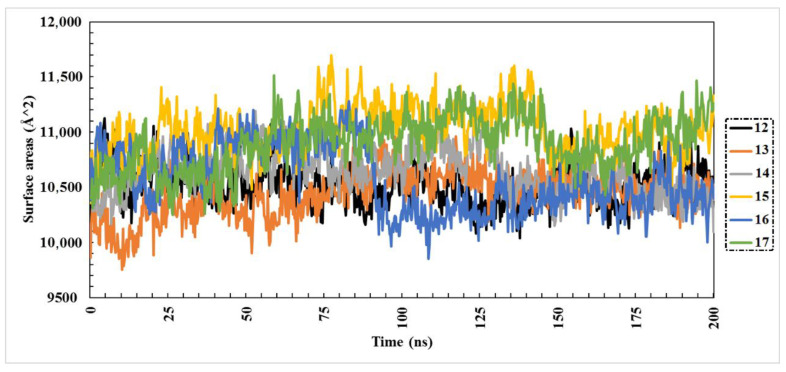
Solvent-accessible surface area (SASA) of the best six compounds for 200 ns MD simulations.

**Table 1 molecules-27-02929-t001:** Docking scores (in kcal/mol) and binding features for best six compounds and remdesivir against Omicron.

Molecule	2D Chemical Structure	Docking Score (kcal/mol)	Binding Features (Hydrogen Bond Length in Å)
**15**	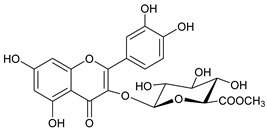	−8.8	ARG403 (3.23 Å), TYR453 (2.94 Å), SER496 (2.99, 3.01 Å), TYR501 (2.92 Å), HIS505 (3.17 Å)
**14**	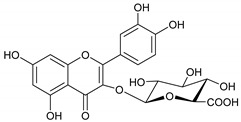	−8.7	ARG403 (3.18 Å), SER496 (2.13, 2.93, 3.01 Å), TYR501 (2.89 Å), HIS505 (3.18 Å)
**16**	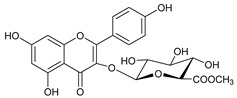	−8.4	GLU406 (2.94 Å), TYR453 (2.87 Å), SER496 (2.92, 3.01 Å), TYR501 (2.90 Å), HIS505 (3.17 Å)
**17**	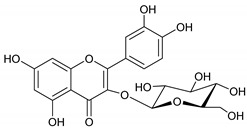	−8.3	TYR453 (2.97 Å), SER496 (2.99, 3.03, 3.08 Å), TYR501 (2.94 Å), HIS505 (3.17 Å)
**13**	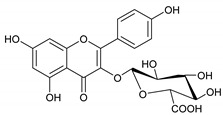	−8.1	TYR453 (2.26 Å), SER496 (2.93, 2.98, 3.01 Å), TYR501 (2.87 Å), HIS505 (3.15 Å)
**Remdesivir**	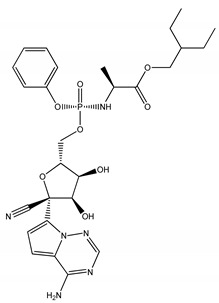	−8.0	TYR453 (2.91 Å), SER494 (2.27, 2.89 Å), TYR501 (3.01 Å)
**12**	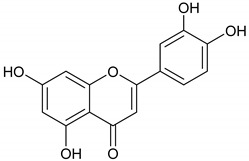	−7.9	ARG403 (3.04 Å), TYR495 (2.57 Å)

**Table 2 molecules-27-02929-t002:** Predicted physiochemical parameters of the best identified compounds and their structural descriptors.

Molecule	miLogP	TPSA	MWt	nON	nOHNH	Nrotb	Nviolations	%ABS
**14**	−0.5	227.6	478.4	13	8	4	2	30.5%
**15**	0.1	216.6	492.4	13	7	5	2	34.3%

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
