# Peer review of "Conducting the RBD of SARS-CoV-2 Omicron Variant with Phytoconstituents from Euphorbia dendroides to Repudiate the Binding of Spike Glycoprotein Using Computational Molecular Search and Simulation Approach"

_molecules, 2022, doi:10.3390/molecules27092929_

Round 1

Reviewer 1 Report

The paper describes the evaluation of 17 compounds isolated by the authors earlier from the plant Euphorbia dendroides for their possible effects on inhibiting the spike protein of the SARS-CoV-2 B.1.1.529 variant. The evaluation was solely done with routine computational methods. Although verifying the results with experiment will provide greater credibility, the paper might still be useful for some readers after the following issues have been addressed.

The experimental details were incomplete and less relevant for this computationally focused paper, they can be removed – citing their previous paper should be sufficient.

The authors might not want to emphasize ΔG=−41.0 kcal/mol to reflect the binding affinity of their best compound predicted by the computational models. MM/PBSA calculations are approximate. It is unclear whether they included the entropic term in the calculations. In addition, the large fluctuations of ΔG obtained from simulations with different lengths shown in Figure 2 suggested large statistical errors in their estimate of this quantity.  Comparing the results with at least one known inhibitor of the protein will provide a useful control (likewise for the docking study).

Equation (1) is incorrect.

Equation (2) does not represent the usual way of calculating root-mean-square fluctuations.

“Solvent-accessible-surface-area (SASA) analysis was conducted to represent the area of the protein subjected to solvent. Consequently, SASA can supply details about the capability of the protein to interact. As seen from data in Figure 6, all six compounds exhibited relative stability and compactness over the 200ns MD simulation course.” These statements are incorrect. As the authors have not convincingly argued the usefulness of presenting these data, they can be removed.

Need to add a reference for molinspiration.

“At a time step of 1.25fs, a regular simulation speed was preserved. With a 298 2fs integration step, all bonds, including hydrogen bonds, were restricted using the 299 SHAKE algorithm [47].”: confusing which time step was actually used.

Times for equilibration before production runs were not specified.

Although clear enough for understanding the science, the English can be improved.

Author Response

Manuscript correction requested (molecules-1688603)

Title: Conducting the RBD of SARS-COV-2 Omicron Variant with Phytoconstituents from Euphorbia dendroides to Repudiate the Binding of Spike Glycoprotein Using Computational Molecular Search and Simulation Approach

Dear Editor-in-Chief

We appreciate the editors' and reviewers' effort and thoughtful remarks on our manuscript. We have implemented their comments and suggestions and would like to submit a revised version of the manuscript to the journal for further consideration. Changes in the initial version of the manuscript are marked up using the “Track Changes” function. Below, we also provide a point-by-point response explaining how we have addressed each of the editors' or reviewers' concerns.

We hope that the revision is acceptable for publication in your journal.

We look forward to the outcome of your evaluation.

Yours sincerely,

On behalf of the co-authors

Ahmed E. Allam

Here is a point-by-point response to the reviewers’ comments and concerns.

Comments from Reviewer 1

** "The paper describes the evaluation of 17 compounds isolated by the authors earlier from the plant Euphorbia dendroides for their possible effects on inhibiting the spike protein of the SARS-CoV-2 B.1.1.529 variant. The evaluation was solely done with routine computational methods. Although verifying the results with experiment will provide greater credibility, the paper might still be useful for some readers after the following issues have been addressed."

Response: Thanks for your appreciation of our work.

** "The authors might not want to emphasize ΔG=−41.0 kcal/mol to reflect the binding affinity of their best compound predicted by the computational models. " 

Response: Thank you for this comment. We have made this change and only highlighted that value in the abstract and conclusion.

** "MM/PBSA calculations are approximate. It is unclear whether they included the entropic term in the calculations."

Response: Thank you for pointing this out. Regarding entropy, we have already included it in the utilized equation in estimating binding energy, and we also added some details in the manuscript for more clarification and to emphasize this point.

** "In addition, the large fluctuations of ΔG obtained from simulations with different lengths shown in Figure 2 suggested large statistical errors in their estimate of this quantity."

Response: Thank you for this comment. Our work was done within YASARA macro called "md_analyzebindenergy". In this script, we just identify our Ligand, and the program estimates the binding energy so there is no chance for any error however we did not perform any statistics. The fluctuation represents the movement of the ligand within the binding cavity over different simulation times (50ns, 100ns, 150ns, and 200ns).

** "Comparing the results with at least one known inhibitor of the protein will provide a useful control (likewise for the docking study)."

Response: "Thank you for this suggestion. We agree with this and have incorporated your suggestion in the molecular docking section."

** "Equation (1) is incorrect."

Response: Thank you for bringing this point to our attention. We have reformulated and corrected it within the manuscript.

** "Equation (2) does not represent the usual way of calculating root-mean-square fluctuations. "

Response: Thank you for bringing this inconsistency to our attention. We have changed it.

** "Solvent-accessible-surface-area (SASA) analysis was conducted to represent the area of the protein subjected to solvent. Consequently, SASA can supply details about the capability of the protein to interact. As seen from data in Figure 6, all six compounds exhibited relative stability and compactness over the 200ns MD simulation course.” These statements are incorrect. As the authors have not convincingly argued the usefulness of presenting these data, they can be removed. "

Response: Thanks for raising this important point. We have made some changes and discussed this type of analysis by providing the average value of SASA for the six structures and explaining their behavior over the 200ns MD simulation.

** "Need to add a reference for molinspiration"

Response: Thank you for this comment. We have included the reference for this tool.

** "At a time step of 1.25fs, a regular simulation speed was preserved. With a 298 2fs integration step, all bonds, including hydrogen bonds, were restricted using the 299 SHAKE algorithm [47].”: confusing which time step was actually used. "

Response: Thank you for pointing this out. This sentence has been modified to clarify the usage of each time step.

** "Times for equilibration before production runs were not specified."

Response: Thank you for mentioning this point. We have included the time for equilibration before production in the MD section.

** Although clear enough for understanding the science, the English can be improved.

Response: Thank you very much for your comments. This manuscript was edited for proper English language, grammar, punctuation, spelling, and overall style by a fluent English speaker. We hope the revised manuscript will meet the requirements of academic publishing in Molecules.

Thank you very much for your fruitful comments, all these notes were covered carefully.

Reviewer 2 Report

The authors have conducted an in silico analysis to find potential inhibitors of the RBD protein of COVID-19. For this, they have used 17 screened compounds from the plant Euphorbia dendroides. Interestingly flavonoids are found to be effective where compound 14 (Quercetin-3-O-β-D-glu- curonopyranoside) and compound 15 (Quercetin-3-O-glucuronide 6''-O-methyl ester) exhibited strong inhibition of Omicron variant. The results are found to be interesting and the article can be accepted after answering minor questions raised below;

  1. The authors said they used NMR for structural elucidation but I don't see any NMR data?
  2. Does the author perform any in vitro functional assay to prove that the compound inhibits RBD protein?

Author Response

Manuscript correction requested (molecules-1688603)

Title: Conducting the RBD of SARS-COV-2 Omicron Variant with Phytoconstituents from Euphorbia dendroides to Repudiate the Binding of Spike Glycoprotein Using Computational Molecular Search and Simulation Approach

Dear Editor-in-Chief

We appreciate the editors' and reviewers' effort and thoughtful remarks on our manuscript. We have implemented their comments and suggestions and would like to submit a revised version of the manuscript to the journal for further consideration. Changes in the initial version of the manuscript are marked up using the “Track Changes” function. Below, we also provide a point-by-point response explaining how we have addressed each of the editors' or reviewers' concerns.

We hope that the revision is acceptable for publication in your journal.

We look forward to the outcome of your evaluation.

Yours sincerely,

On behalf of the co-authors

Ahmed E. Allam

Here is a point-by-point response to the reviewers’ comments and concerns.

Comments from Reviewer 2

The authors have conducted an in silico analysis to find potential inhibitors of the RBD protein of COVID-19. For this, they have used 17 screened compounds from the plant Euphorbia dendroides. Interestingly flavonoids are found to be effective where compound 14 (Quercetin-3-O-β-D-glu- curonopyranoside) and compound 15 (Quercetin-3-O-glucuronide 6''-O-methyl ester) exhibited strong inhibition of Omicron variant. The results are found to be interesting and the article can be accepted after answering minor questions raised below;

Response: Thanks for your appreciation of our work.

  1. The authors said they used NMR for structural elucidation but I don't see any NMR data?

Response: NMR spectra of listed compounds have been added in the supplementary file (Figures S1-S52).

  1. Does the author perform any in vitro functional assay to prove that the compound inhibits RBD protein?

Response: This step is being planned and will be further taken into consideration in near future.

Round 2

Reviewer 1 Report

The manuscript has been improved.